# Matrix Metalloproteinase-9 Level in Synovial Fluid—Association with Joint Destruction in Early Rheumatoid Arthritis

**DOI:** 10.3390/medicina59010167

**Published:** 2023-01-14

**Authors:** Sonja K. Stojanovic, Bojana N. Stamenkovic, Jovana M. Cvetkovic, Valentina G. Zivkovic, Marija R. Andjelkovic Apostolovic

**Affiliations:** 1Faculty of Medicine, University of Nis, 18000 Nis, Serbia; 2Institute for Treatment and Rehabilitation “Niska Banja”, 18205 Niska Banja, Serbia; 3Public Health Institute Nis, 18000 Nis, Serbia

**Keywords:** rheumatoid arthritis, matrix metalloproteinase-9, radiographic progression

## Abstract

*Background and objective*: Matrix metalloproteinases (MMPs) are the key enzymes in the pathogenesis of cartilage and joint damage and potentially a new biomarker of the early erosive form of rheumatoid arthritis (RA). Firstly, the study aimed to compare the level of MMP-9 in plasma (PL) and synovial fluid (SF) of patients with RA and osteoarthritis (OA). Secondly, the goal was to examine the association of MMP-9 level in PL and SF with early erosive changes in RA, and finally, to determine the association of MMP-9 level with serological parameters of the disease (rheumatoid factor-RF and anti-citrulline protein antibodies-ACPA). *Materials and Methods*: A total of 156 subjects were involved in this study (84 patients with RA and 72 patients with OA, who were involved as a control group). MMP-9 level was measured in PL and SF of all subjects by the sandwich enzyme-linked immunosorbent assay (ELISA) method. Standard radiographs of the hands and feet were used to detect joint damage and classification into erosive or non-erosive RA. The Larsen score (LS) was used for the quantitative assessment of joint damage, and its annual change (∆ LS) was used to assess the radiographic progression of the disease. *Results:* MMP-9 level in PL and SF was significantly higher in RA compared to controls (PL: 19.26 ± 7.54 vs. 14.57 ± 3.11 ng/mL, *p*< 0.01; SF: 16.17 ± 12.25 vs. 0.75 ± 0.53 ng/mL, *p* < 0.001) as well as in SF of patients with erosive compared to non-erosive RA (18.43 ± 12.87 vs. 9.36 ± 7.72; *p* < 0.05). Faster radiographic progression was recorded in erosive compared to non-erosive early RA (11.14 ± 4.75 vs. 6.13 ± 2.72; *p* < 0.01). MMP-9 level in SF, but not in PL, significantly correlates with the radiographic progression in both erosive and non-erosive RA (ρ = 0.38 and ρ = 0.27). We did not find a significant association between RF and MMP-9 level in early RA, but the ACPA level significantly correlates with MMP-9 level in SF (r = 0.48). *Conclusion:* The level of MMP-9 in plasma and synovial fluid of patients with RA is significantly higher compared to patients with osteoarthritis. The level of MMP-9 in synovial fluid is significantly higher in erosive than non-erosive early RA. It is significantly associated with the radiographic progression of the disease and the level of anti-citrulline protein antibodies.

## 1. Introduction

Rheumatoid arthritis (RA) is a chronic inflammatory disease caused by the autoimmune destruction of synovial joints. If it has a progressive course and is inadequately treated, it can result in significant disability and early mortality. The main goals of RA treatment are controlling inflammation, preventing joint damage, and preserving functional ability.

RA differs from other inflammatory arthropathies in its pronounced tendency towards joint destruction.

Cartilage and subchondral bone degradation in arthritis are mediated by different mechanisms and involve various cells. Synoviocytes, especially fibroblast-like synoviocytes, attack and damage the cartilage in the affected joint, while bone destruction is mediated by osteoclasts activated following the release of pro-inflammatory cytokines produced by the synovium. Proteolytic enzymes actively participate in both of these processes (chondral and bone breakdown), whose activity is increased at the location of joint destruction [1].

Matrix metalloproteinases (MMPs) are a family of Zn^2+^-dependent endopeptidases which actively participate in the destruction of extracellular matrix components. They have been shown to have an important role in the invasion of the synovial tissue in cartilage, cartilage destruction, and bone erosion formation [2]. In the joint capsule, it is produced by monocytes and macrophages; production by chondrocytes appears minimal. Tissue inhibitor MMP (TIMP) prevent uncontrolled degradation of ECM components. The increased proteolytic activity of MMPs is due to an imbalance between proteolytic and inhibitory activities of MMPs and TIMP [3].

Gelatinases, also known as MMP-2 and MMP-9, are the best characterised and investigated type of MMPs, expressed in joint tissues of patients with RA and OA [4]. The results of previous studies suggest that MMP-9 levels in SF were higher in patients with rheumatoid arthritis compared to osteoarthritis [5]. Compared with OA synovial fibroblasts and normal synovial fibroblasts, which produce only MMP-2, a unique feature of RA synovial fibroblasts is the production of both MMP-2 and MMP-9 [6,7].

According to literature data, both MMP-9 and MMP-2 contribute to the survival, proliferation, migration and invasion of RA synovial fibroblasts, with MMP-9 having stronger effects, while MMP-2 inhibits these parameters [2]. In addition to breaking down denatured collagen, MMP-9 activates other MMPs primarily pro MMP-13 collagenase, the main enzyme involved in cartilage erosion in both RA and OA [8,9]. Based on these results, it is hypothesised that selective blockade of MMP-9 may reduce inflammation and cartilage damage in RA.

The significance of MMP-9 for RA prognosis, particularly if they are determined in the early stages of the disease when we still cannot detect joint damage with the visualisation methods at our disposal, as well as the relationship between locally released and circulating MMPs and serological markers of RA (rheumatoid factor-RF and anti-citrulline protein antibodies-ACPA) have not been fully clarified yet.

Destruction of articular cartilage is an early phenomenon in arthritis; its control today is one of the therapeutic goals of RA. In this sense, it is imperative to predict which profile of RA patients tends to rapid joint destruction to stop the progression of structural joint damage with early aggressive treatment.

Until recently, decision making on RA treatment mainly relied on the assessment of disease activity, and therapy adjustment was executed according to the “treat to target” (T2T) principle, the goal of which is remission or low disease activity [10]. However, there is a limitation of this strategy concerning the prevention of structural joint changes. Namely, radiographic progression has been reported in some patients with RA in remission, indicating the limitation of the underlying disease activity indicator for predicting radiographic progression [11]. That is why new biomarkers (inflammatory cytokines, proteinases, collagen breakdown products...) are increasingly being advocated in daily practice to assess the radiological progression of RA [12].

Few studies have examined the predictive significance of MMP-9 expression in plasma and synovial fluid for the early onset of erosive RA [13,14]. Therefore, examining the association of MMP-9 levels, especially in synovial fluid, with early destructive joint changes and the serological profile of RA patients is essential as a potential indicator of developing a rapidly progressive form of the disease.

The main objective of the study was to determine the association of MMP-9 level in plasma and synovial fluid with early erosive changes in RA, as well as the association of MMP-9 level with immunoserological parameters of the disease (RF and ACPA). We also compared the level of this endopeptidase in PL and SF in patients with RA and OA.

## 2. Materials and Methods

### 2.1. Study Design

The prospective comparative study was conducted at the Clinic for Rheumatology of the Institute for Treatment and Rehabilitation “Niska Banja” from 2017 to 2020 and included 156 subjects. Inclusion criteria were primary diagnosed RA or OA. All included patients had knee synovitis at the beginning of the study and were not treated with intra-articular injections of steroids or chondroprotectors for at least three months before this study. Patients underwent knee aspiration for diagnostic or therapeutic arthrocentesis.

The diagnosis of RA was made based on the ACR/EULAR criteria from 2010 [15] and patients with OA met the EULAR classification criteria for osteoarthritis of the knee [16]. Exclusion criteria were the diagnosis of autoimmune diseases other than RA, use of glucocorticoids, use biological disease-modifying anti-rheumatic drugs or JAK (janus kinase) inhibitors, acute and chronic infections, malignant diseases, or severe lung, liver, kidney or endocrinological diseases, younger than 18 and older than 80. Patients with OA did not have an acute knee injury.

The RA patient group included 84 patients with early RA with the duration of symptoms 6 months or less. Seventy-two patients with osteoarthritis were included in the study as a control group (CG). The RA group (men/women 28/56) included patients of an average age of 60.59 ± 11.29 years. All patients with RA included in the study were treated with methotrexate (average weekly dose 15.75 ± 4.76 mg) with a minimum duration of therapy of one year. After 3–6 months, the treatment escalated to the disease activity in 32 (38.01%) patients of the RA group. Patients from the CG were comparable in demographic characteristics to the RA group (men/women: 20/52, average age 62.70 ± 6.18). 

First, we compared the level of MMP-9 in PL and SF of RA and OA patients. For further analysis the whole RA group was divided into two subgroups (erosive and non-erosive RA according to joint damage on hands and feet radiographs at the beginning of the study). We compared the level of MMP-9 in PL and SF of patients with erosive and non-erosive RA, the degree of joint destruction measured by Larsen score (LS), and the annual progression of LS in these two subgroups of RA patients. In the studied RA group, we examined the association of MMP-9 level with serological parameters of the disease (rheumatoid factor-RF and anti-citrulline protein antibodies-ACPA).

All participants in the study consented to participate by signing the informed consent. The Ethics Committee approved the study of the Faculty of Medicine of the University of Nis (number 01-206-6) and the Ethics Committee of the “Niska Banja” Institute (number 03-7974). The study was carried out following the Helsinki Declaration on Human Clinical Research. Research is part of a scientific project of the Ministry of Education, Science and Technological Development of the Republic of Serbia [III41018].

### 2.2. Measurements

#### 2.2.1. MMP–9 Measurement

Plasma was prepared after blood collection and all samples were stored at −20 °C before analysis. SF was aspirated from the knee joints of patients with RA or OA under aseptic conditions and stored at −80 °C before being analysed. 

MMP-9 was investigated in the plasma and synovial fluid of each patient. The total MMP-9 level was determined using the sandwich enzyme-linked immunosorbent assay (ELISA; Amersham Biosciences, Little Chalfont, UK) according to the manufacturer’s instructions. Absorbance was detected at 450 nm. Enzyme level in each sample was determined by interpolation from the standard curve and expressed in ng/mL. The sensitivity of the assay was 0.6 ng/mL. The intra-assay and inter-assay CV were 2.3% and 7.5%, respectively. The minimum detectable dose was less than 0.156 ng/mL and there was no significant cross-reactivity or interference with other proteins.

#### 2.2.2. Laboratory Analyses

The following serological tests were performed on all RA patients: RF and ACPA. The tests were performed in a certified local laboratory. The presence and concentration of IgM RF were determined by the latex agglutination method (Huma Tex RF “Human”—Germany). Results with RF concentration higher than 20 IU/L were considered positive. The presence of anti-CCP antibodies (ACPA) was determined by the ELISA (enzyme-linked immunosorbent assay) method (Imtec-CCP antibodies “Human”—Germany), which determines the IgG class of antibodies against citrullinated peptides. Samples were classified as positive in case of values >25 U/mL.

#### 2.2.3. Radiography of Joints

Radiographs of the hand and feet were taken at baseline and after 12 months and were consensus-scored by two trained readers using the modified Larsen score. The value range of the modified Larsen score was 0 to 200. The analysis included radiographic evaluation of 32 joints of hands and feet (eight proximal interphalangeal joints of hands, two interphalangeal joints of hand thumbs, ten metacarpophalangeal joints, two radiocarpal joints, two interphalangeal joints of hallux and eight metatarsophalangeal joints). After comparison with the standard referent ranges, changes in the joints were graded from 0–5: grade 0—intact joint space; grade 1—soft tissue swelling, joint space narrowing and periarticular osteoporosis; grade 2—erosions and joint surface destruction up to 25%; grade 3—erosions and joint surface destruction of 26–50%; grade 4—erosions and joint surface destruction of 51–75%; grade 5—erosions and joint surface destruction above 75% [17]. The patients of the study group were divided into those with erosive and non-erosive RA concerning the presence of destructive changes (cysts and erosion) found during the initial radiographs. Radiographic progression during the first year of RA was assessed based on the annual change in the Larsen score.

##### Statistical Analysis

Data were analysed using SPSS software (Inc. Armonk, NI, USA, version 22.0). The normality of data was analysed with the Kolmogorov–Smirnov test. Data are presented as mean ± standard deviations with a 95% confidence interval. Normally distributed data were compared by Student t-test, while the Mann–Whitney and Wilcoxon signed-rank tests were used for abnormally distributed data. The correlation was determined by Pearson and Spearman rank correlation. Univariate logistic regression analyses with an odds ratio (OR) were used to assess specific risk factors’ impact on erosion development. The *p*-value < 0.05 was considered statistically significant.

## 3. Results

The MMP-9 level in the plasma and synovial fluid of patients with RA is significantly statistically higher than the level of this enzyme in the plasma and synovial fluid of patients from the control group. A statistically significant difference (*p* < 0.001) was found for MMP-9 level in synovial fluid (16.17 ± 12.25 vs. 0.75 ± 0.53) concerning the level of this enzyme in plasma (19.26 ± 7.54 vs. 14.57 ± 3.11) (*p* < 0.01) (Table 1).

There was no statistically significant difference in the patients with erosive and non-erosive RA groups concerning gender, age, and disease duration. In the RA group, most patients (48–57.14%) had radiologically verified erosive joint changes at the beginning of follow-up. Non-erosive arthritis was found in 36 (42.86%) patients.

Table 2 shows the level of MMP-9 in plasma and synovial fluid in RA patients with and without erosions at the beginning of the study. MMP-9 level in plasma is higher in erosive arthritis than in non-erosive arthritis (21.37 ± 10.50 vs. 18.62 ± 6.37) with no statistical significance. The level of this enzyme in synovial fluid in erosive arthritis is significantly statistically higher than the level of MMP-9 in patients with non-erosive RA (18.43 ± 12.87 vs. 9.36 ± 7.72), *p* < 0.05.

The logistic regression model has shown four predictors for erosion development (age, RA duration, MMP-9 level in blood plasma and MMP-9 level in synovial fluid). Of all those four predictors for erosive RA—age, RA duration and MMP-9 level in synovial fluid proved to be statistically significant. The most significant predictor of the development of erosion in early RA is the level of MMP-9 in synovial fluid. An increase in MMP-9 level in synovial fluid by one measurement unit increases the likelihood of erosion almost three-fold (OR: 2.8) (Table 3).

Using the modified Larsen score, we quantitatively assessed the degree of destructive joint changes in the examined RA group, both before the start of MTX therapy and after a year of treatment. As expected, patients with an initially erosive form of the disease had a significantly higher value of the Larsen score at the beginning and after 12 months of therapy. Annual progression of the Larsen score is significantly higher in erosive compared to non-erosive early RA (11.14 ± 4.75 vs. 6.13 ± 2.72; *p* < 0.01) (Table 4).

The association of MMP-9 level in the plasma and synovial fluid of the RA group with the progression of structural joint changes (annual change in the Larsen score) was determined by Spearman’s rank correlation coefficient (ρ). We found a highly significant correlation of MMP-9 level in synovial fluid in both erosive (ρ = 0.38) and non-erosive RA (ρ = 0.27) (*p* < 0.01) (Figure 1 and Figure 2). The level of this enzyme in plasma did not significantly correlate with the annual change of LS in early RA (ρ = 0.16 for erosive and ρ= 0.13 for non-erosive RA).

The serological profile of the studied RA group involved testing the concentration of rheumatoid factor (RF) and antibodies to citrullinated polypeptides (ACPA). Examining the correlation of MMP-9 level in plasma and synovial fluid of the RA group, we determined that MMP-9 level in synovial fluid was statistically highly correlated with ACPA in serum (*p* < 0.01) (Table 5).

## 4. Discussion

Bone and cartilage destruction is an irreversible process causing the loss of joint function in both RA and OA. One of the aims of our study was to quantify the level of MMP-9 in PL and SF of patients with RA compared to patients with OA. According to data from the literature, patients with knee OA have a pro-inflammatory SF environment, similar to RA patients, that may contribute to ongoing cartilage loss and ECM degradation [18]. By comparing 16 biomarkers in the SF of patients with OA and RA, the authors showed that the level of only six biomarkers was significantly higher in SF than active RA compared to OA: TNF-α, IL-1-β IL-7, MMP-1, MMP-2 and MMP-3. The level of MMP- 9 in SF did not differ in patients with OA and RA [19]. 

According to our results, MMP-9 level is higher in patients with RA than in patients with osteoarthritis, both in plasma (19.26 ± 7.54 vs. 14.57 ± 3.11 ng/mL) and synovial fluid (16.17 ± 12.25 vs. 0.75 ± 0.53 ng/mL). A more significant difference was noted regarding this enzyme’s level in the subjects’ synovial fluid. This was expected, given that the synovial fluid in the affected joint is a medium in which mediators of the bone/cartilage destruction process are present. Similar to our results, a group of Japanese authors suggested a significantly higher level of MMP-9 in the synovial fluid of RA patients compared to people with osteoarthritis [20]. 

For the control group in our research, we had patients with degenerative rheumatism of the knee joints. It is well-established that osteoarthritis is also a consequence of cartilage and subchondral bone damage and that metalloproteinases participate in this process. In our CG with knee OA, the level of MMP-9 in SF was low. We showed that the level of MMP-9 in the plasma of OA patients was higher than the level of MMP-9 in SF, similar to the results of authors from the USA [21]. However, Tchetverikov showed in earlier works that the level of MMP-9 in the plasma and synovial fluid of patients with osteoarthritis is slightly increased, and there is no significant difference between the level of MMP-9 in the circulation and the level of MMP-9 in the joint fluid [22,23]. Therefore, the measurement of MMP-9 is unlikely to be an accurate biomarker that reflects the inflammatory process in patients with knee OA.

Based on the elevated local level of MMPs in the synovial fluid of RA patients, some authors considered the local application of tissue MMP inhibitors as a potential therapeutic strategy to slow down structural damage in RA [24]. 

In this study, we followed the radiological progression of early RA with the T2T treatment concept. We examined the significance of the basal level of MMP-9 for the development of early erosions and the progression of structural changes in RA, as well as the association of the level of this enzyme with the serological parameters of disease activity (RF and ACPA).

Increased level of MMP-9 is associated with the early erosive form of RA. Patients with RA who had erosive joint changes at the beginning of our study had significantly higher MMP-9 enzyme levels in the synovial fluid than patients without erosive joint changes (18.43 ± 12.87 ng/mL versus 9.36 ± 7.72 ng/mL), *p* < 0.05. This fact directly highlights the importance of MMP-9 in the process of cartilage and bone damage, characteristic of RA. The results of our research can build up on the results of previously published studies that indicate the role of MMPs in the formation of erosions and highlight their predictive importance for the progression of joint damage [25]. In addition to the MMP-9 level, our earlier works [26] examined the importance of genetic factors (TNF alpha gene polymorphism) for the radiological progression of RA using the multivariate regression method. Similar to the previous results, here we concluded that the patient’s age independently influences the development of erosive arthritis, the disease’s duration and the level of MMP-9 in the synovial fluid, which is coherent with the results of other researchers [27].

Assessment of the risk of rapid radiographic progression is essential for the prognosis of RA. We recorded more significant radiographic progression, expressed by the annual progression of the Larsen score, in the group of patients with initially erosive arthritis compared to non-erosive arthritis (11.14 ± 4.75 vs. 6.13 ± 2.72; *p* < 0.01). Similar to these results, our earlier research indicates that the Larsen score at the beginning of the disease has the most significant predictive value for joint damage after four years of RA [28]. A multicentre retrospective study in Hungary also highlighted the presence of erosions at the onset of the disease as a significant risk factor for the rapid radiographic progression of RA [29]. Data analysis from the DREAM (Dutch Rheumatoid Arthritis Monitoring) registry provides valuable information on predictors of radiological damage during 6 years of treatment with the T2T concept. The authors concluded that initial erosive changes and ACPA positivity are associated with long-term radiological disease progression [30,31].

Following the available results from the literature [32], the predictive significance of MMP-9 level for the rapid progression of RA has been confirmed by our results of a significant correlation of the level of this enzyme in both erosive and non-erosive early RA. 

Stopping the progression of joint damage is the imperative of all types of RA treatment today. Nowadays, the RA treatment concept implies early diagnosis before structural joint damage, with early aggressive therapy, especially in people with predisposing risk factors for rapid radiographic progression. Understanding these predictive factors may be necessary for the initial therapeutic approach and rapid escalation of the therapeutic regime to slow down structural joint damage. Identifying SF biomarker profiles for different phenotypes of patients with RA can improve precise therapeutic strategies based on the principle of a personalised therapeutic approach.

Our results confirmed the importance of MMP-9 level in the synovial fluid of patients with RA for developing early erosive arthritis. However, the determination of the concentration of this enzyme is still not routine; therefore, we examined the association of MMP-9 level with the standard serological profile of the disease (RF and ACPA). Regarding the progression of the disease, ACPA positivity was associated with a more severe, erosive phenotype of RA. Numerous studies have confirmed the association of ACPA with rapid joint destruction, while the contribution of RF concerning ACPA has diminished [33]. Like the results of these studies, our results highlight a significant correlation of ACPA with MMP-9 level in synovial fluid, highlighting the greater importance of this serological parameter than RF for the rapid radiological progression of early RA. According to our results, multivariable risk assessment of rapid radiological progression of early RA showed that ACPA positivity with a polygenic risk score, as well as female gender and body mass index was associated with rapid progression of structural joint changes [34]. 

The treatment of patients with rheumatoid arthritis (RA) has not been personalised enough yet, as there are no reliable predictors of the course of the disease. Future studies will reveal whether adding and combining different biomarkers, including MMP-9 level in synovial fluid, will increase the accuracy of risk models predicting the radiographic progression of RA.

This study has several potential limitations. The number of patients with early RA was too small to provide validated evidence regarding radiographic progression. We did not investigate the effects of disease activity on MMP-9 level and joint damage. We did not analyse the biochemical profile of SF as a predictor of the radiographic progression of RA, as well as the level of other MMPs, primarily MMP-2. Additional studies should be performed to support the present conclusions. Future longitudinal studies are needed to confirm these results.

## 5. Conclusions

The MMP-9 level in the plasma and synovial fluid of patients with RA is higher than this enzyme’s level in patients with osteoarthritis. The level of MMP-9 in the synovial fluid is significantly higher in initially erosive compared to non-erosive early RA. It is significantly associated with the radiographic progression of the disease and the level of anti-citrulline protein antibodies. Our results suggest that level of MMP-9 enzyme in the synovial fluid of patients with early RA can predict the rapid radiographic progression of the disease. Future studies will reveal whether adding and combining different biomarkers will increase the accuracy of risk models predicting radiographic progression in RA.

## Figures and Tables

**Figure 1 medicina-59-00167-f001:**
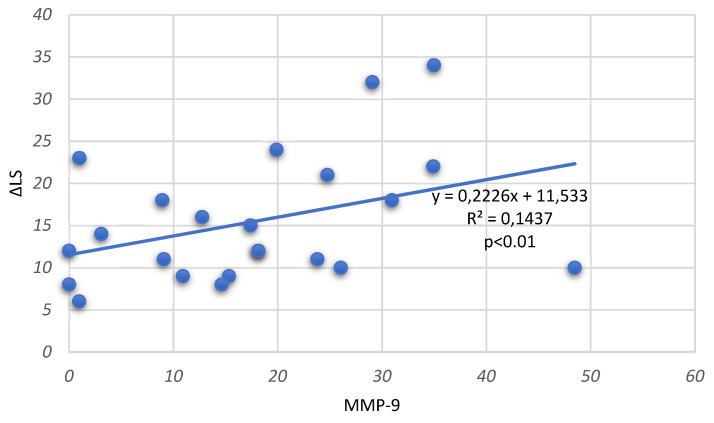
Correlation of MMP-9 in synovial fluid with annual changes of Larsen score (ΔLS) in patients with erosive RA.

**Figure 2 medicina-59-00167-f002:**
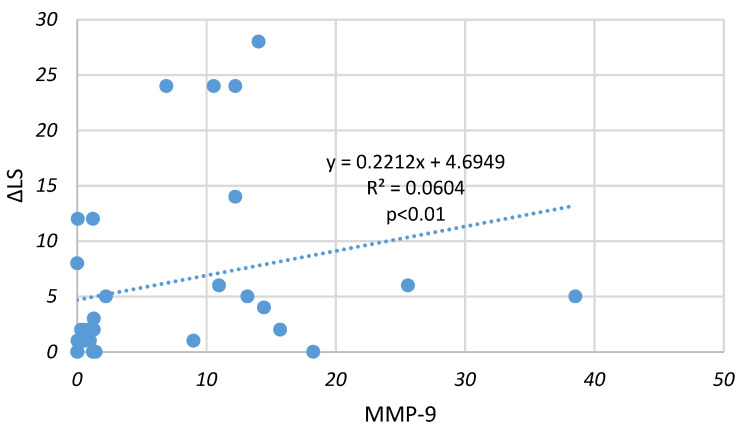
Correlation of MMP-9 in synovial fluid with annual changes of Larsen score (ΔLS) in patients with non-erosive RA.

**Table 1 medicina-59-00167-t001:** MMP-9 level in plasma and synovial fluid in RA and control group patients.

	Group	Mean ± SD (ng/mL)	95% CI	Median	CV
Plasma	RA (n = 84)	19.26 ± 7.54 **	17.5–20.5	19.76	41.56
	Controls (n = 72)	14.57 ± 3.11	13.3–14.7	14.57	21.57
Synovial fluid	RA (n = 84)	16.17 ± 12.25 ***	13.4–18.6	14.21	84.82
	Controls (n = 72)	0.75 ± 0.53	0.74–0.76	0.82	62.68

RA—rheumatoid arthritis; SD—standard deviation; n—number of subjects; CV—coefficient of variation; 95% CI—confidence interval for mean; ** *p* < 0.01; *** *p* < 0.001.

**Table 2 medicina-59-00167-t002:** MMP-9 level (ng/mL) in plasma and synovial fluid of patients with erosive and non-erosive RA.

MMP-9 Level	Erosive RA (n = 48)	Non-Erosive RA (n = 36)
Plasma (ng/mL) (mean ± SD)	21.37 ± 10.50	18.62 ± 6.37
Synovial fluid (ng/mL)(mean ± SD)	18.43 ± 12.87 *	9.36 ± 7.72

RA—rheumatoid arthritis; SD—standard deviation; n—number of subjects; * *p* < 0.05; data are presented as mean ± SD.

**Table 3 medicina-59-00167-t003:** Logistic regression model for the assessment of certain risk factors’ impact on the erosion development.

Factor	OR	95% CI	*p*
Age	1.11	1.02–1.22	0.019
RA duration	1.16	1.01–1.32	0.030
MMP-9 level in plasma	0.95	0.85–1.06	0.330
MMP-9 level in synovial fluid	2.80	1.24–6.30	0.012

OR—odds ratio; 95% CI—confidence interval; *p*—*p*-value; MMP-9—matrix metalloproteinase-9; RA—rheumatoid arthritis.

**Table 4 medicina-59-00167-t004:** Larsen score in patients with RA according to the presence of erosions on radiographs.

Larsen Score	Erosive RA (n = 48)	Non-Erosive RA (n = 36)
Before MTX therapy (mean ± SD)	48.58 ± 20.84	18.78 ± 12.91 *
After 12 months of MTX therapy (mean ± SD)	65.32 ± 22.24	26.48 ± 13.27 ***
∆LS (mean ± SD)	11.14 ± 4.75	6.13 ± 2.72 **

n—number of subjects; MTX—methotrexate; SD—standard deviation; ∆LS—annual changes in Larsen score; data are presented as mean ± SD; * *p* < 0.05; ** *p* < 0.01; *** *p* < 0.001.

**Table 5 medicina-59-00167-t005:** Correlation of MMP 9 level in plasma and synovial fluid with serological parameters of RA.

Parameter	RF	ACPA
MMP-9 level in plasma (ng/mL)	−0.20	0.18
MMP-9 level in synovial fluid (ng/mL)	−0.06	0.48 **

MMP-9—matrix metalloproteinase-9; RA—rheumatoid arthritis; RF—rheumatoid factor; ACPA—anti-citrullinated protein antibody; ** *p* < 0.01.

## Data Availability

The data presented in this study are available on request from the corresponding author.

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
