# Peer review of "Matrix Metalloproteinase-9 Level in Synovial Fluid—Association with Joint Destruction in Early Rheumatoid Arthritis"

_medicina, 2023, doi:10.3390/medicina59010167_

Round 1

Reviewer 1 Report

This study revealed the correlation between MMP-9 activity in RA synovial fluid and radiographic progression, which suggested that MMP-9 could be a diagnostic marker for erosive RA. Although the concept is important in clinical practice, I have the following issues to be addressed,

1. Throughout the whole manuscript, please make sure the term" MMP-9" activity. After reading the method, they used ELISA to detect MMP-9 in each sample. Please be sure that ELISA is used to detect protein expression, not activity. As published previously (Clin Exp Rheumatol.2004 May-Jun;22(3):335-8), gelatin zymography is the method to evaluate MMP-9 or -2 activity. I suggested they had to make sure the term "MMP-9 activity" or " MMP-9 level".

2. As per the reference I have provided above, both MMP-2 and -9 activities are increased in RA synovial fluid, and the 2 enzymes are the gelatinase. Please explain why choosing only  MMP-9 in this study.

3. On p5, lines 185-191, please provide p values of fig1 and 2, and the data regarding "The activity of this enzyme in plasma did not significantly correlate with the annual change of LS in early RA (please include p-value)".  

Reviewer 2 Report

2. Materials and Methods

2.1. Study Design

you describe their study population:

“[…]84 patients with early RA […] The diagnosis of RA was made within six months of the onset of symptoms […] All patients with RA included in the study were treated with methotrexate (average weekly dose 15.75±4.76 87 mg) with a minimum duration of therapy of one year. […]”

1)     Do you need the control group for the objective of the study?

2)     Should they collect in Study Design the erosive and non erosive RA subgroups?

3)     you don't indicate how long it took to collect the study sample?

4)     you don't indicate if you have calculated the sample size of the study?

5)     How much time did you spend on the study?

[…] Included 156 subjects with knee synovitis: […]

6)     your inclusion criterion is: “ […] Included 156 subjects with knee synovitis: […]”

7)     Why then use the modified Larsen evaluation method if the knee is not included in it?

8)     How is it related to RA? Is it the only clinical presentation of the patients?

80-90

4. Discussion

“For the control group in our research, we had patients with […] These results differ from a recent study, which did not show a higher activity of MMP-9 in the plasma of patients with RA compared to healthy subjects.”

9)     What is the relationship to your objective?

231-239

10)  Limitations have you found in your study?

In general, when reviewing the study, it gives the impression that they are two different studies, in their approach and study population that define AR and CG, and their results focus on the difference between the two subgroups (Erosive AR and Non-erosive AR) that they create without reference to their study population.

Round 2

Reviewer 2 Report

Nice job.